# *Gurltia paralysans*: A Neglected Angio-Neurotropic Parasite of Domestic Cats (*Felis catus*) and Free-Ranging Wild Felids (*Leopardus* spp.) in South America

**DOI:** 10.3390/pathogens11070792

**Published:** 2022-07-13

**Authors:** Lisbeth Rojas-Barón, Anja Taubert, Carlos Hermosilla, Marcelo Gómez, Manuel Moroni, Pamela Muñoz

**Affiliations:** 1Biomedical Research Center Seltersberg (BFS), Institute of Parasitology, Justus Liebig University Giessen, 35392 Giessen, Germany; anja.taubert@vetmed.uni-giessen.de (A.T.); carlos.r.hermosilla@vetmed.uni-giessen.de (C.H.); 2Facultad de Ciencias Veterinarias, Instituto de Farmacología y Morfofisiología, Universidad Austral de Chile, Valdivia 5090000, Chile; marcelogomez@uach.cl; 3Facultad de Ciencias Veterinarias, Instituto de Patología Animal, Universidad Austral de Chile, Valdivia 5090000, Chile; manuelmoroni@uach.cl (M.M.); pamela.munoz@uach.cl (P.M.)

**Keywords:** *Gurltia paralysans*, life cycle, angio-neurotropic nematode, Metastrongyloidea, domestic cats, wild felids

## Abstract

*Gurltia paralysans* is a neglected and re-emerging metastrongyloid angio-neurotropic nematode causing severe chronic meningomyelitis in domestic cats (*Felis catus*) as well as in free-ranging small wild felids such as kodkods (*Leopardus guigna*), margays (*Leopardus wiedii*) and the northern tiger cat (*Leopardus triginus*) in South America. Within these definitive hosts (DH), adult males and females of *G. paralysans* parasitize the leptomeningeal veins of the subarachnoid space and/or the meningeal veins of spinal cord parenchyma, inducing vascular alterations. Feline gurltiosis has been associated with progressive thrombophlebitis of the meningeal veins, resulting in ambulatory paraparesis, paraplegia, ataxia, hindlimb proprioceptive deficit, uni- or bilateral hyperactive patellar reflexes, faecal and urinary incontinence, and tail paralysis. The complete life cycle of *G. paralysans* has not been elucidated yet, but most probably involves gastropods as obligate intermediate hosts (IH). In terms of epidemiology, *G. paralysans* infections in domestic and wild felids are scattered around various South American countries, with hyperendemic areas in southern parts of Chile. Etiological diagnosis of *G. paralysans* still represents a challenge for clinicians due to a lack of evidence of the excretion of either eggs or larvae in faeces or in other body fluids. Diagnosis is based on clinical neurological signs, imaging findings through computed tomography (CT), myelography, magnetic resonance imaging (MRI), and post mortem examination. Nonetheless, novel diagnostic tools have been developed, including semi-nested PCR for detecting circulating *G. paralysans* DNA in the cerebrospinal fluid, serum and blood samples as well as in serological diagnostic kits detecting parasite-derived antigens, but these need validation for routine usage. The hypothetical life cycle of *G. paralysans* is addressed in this article, including the exogenous stages (i.e., eggs, and first- (L1), second- (L2) and third-stage (L3) larvae) and obligate gastropod IH and/or paratenic hosts (PH), and we propose possible anatomical migration routes of infective L3 that reach the leptomeningeal veins in vivo. Finally, the pro-inflammatory endothelium- and leukocyte-derived innate immune reactions of the host against *G. paralysans*, which most likely result in thrombophlebitis and meningomyelitis, are briefly touched on.

## 1. Introduction

*Gurltia paralysans* is a neglected metastrongyloid nematode (superfamily Metastrongyloidea, family Angiostrongylidae) causing severe chronic meningomyelitis in domestic cats (*Felis catus*) as well as in free-ranging wild felids of the genus *Leopardus* in South America. Historically, *G. paralysans* was first described by Kurt Wolfgang Wolffhügel (1933), a German scientist, naturalist, and parasitologist, who isolated adult nematodes from the vein system of the leptomeninges of 11 domestic cats suffering from chronic pelvic paraparesis within the provinces of Llanquihue and Puerto Varas in Southern Chile [1,2]. The etymology of the genus *Gurltia* is explained by Wolffhügel’s intention to honour Ernst Friedrich Gurlt (1794–1882), a German veterinary anatomist and teratologist. Wolffhügel (1933) initially described this neurological parasitosis associated with small felines as “*paraplejia cruralis parasitaria felis*” and placed the nematode species within the genus *Hemostrongylus*, later renamed *Angiostrongylus* [1]. One year later, Wolffhügel (1934) published an extended description of its geographic distribution, morphology, pathological findings, and clinical signs and speculated on its transmission and definitive host (DH) spectrum. This author showed the small wild felid species kodkod (*Leopardus guigna;*
Figure 1) to be the main natural DH in Southern Chile and in the border regions of Argentina, locally known as “guiña” or “spotted tiger cat”. In addition, Wolffhügel (1993, 1934) proposed domestic cats as aberrant DH that were first introduced by European settlers into the South American continent and thereafter became exposed to this endemic nematode [3]. More recently, the spectrum of wild felid species acting as DH has increased, nowadays including the margay (*Leopardus wiedii*) and the northern tiger cat (*Leopardus tigrinus*) [4]. Additionally, within the genus *Leopardus*, other small wild felids have also been suggested as potential DH in South America [1,2,3,4,5,6,7]. Alongside the genus *Leopardus*, larger wild felids of South America, i.e., pumas (*Puma concolor concolor*), jaguars (*Panthera onca*), and jaguarondis (*Herpailurus yagouaroundi*), have also been suggested as potential DH, but this needs further investigation [1,5,7]. In a recent study on free-ranging guiñas in Chile, although no presence of *G. paralysans* was observed, the isolation of other closely related nematodes such as *Angiostrongylus* sp., *Oslerus* sp. and *Troglostrongylus* sp. was observed, indicating diversity, susceptibility, and the potential risk of lungworm infections in South America [8].

In vivo, *G. paralysans* has a marked angio-neurotropism invading the venous system of leptomeninges, specifically the thoracic, lumbar, and sacral spinal cord segments. The distribution of *G. paralysans* adults within the meningeal veins of the subarachnoid space implies activation of the highly immunoreactive endothelium, as seen for *Angiostrongylus vasorum* [9], probably resulting in thrombophlebitis with thrombus formation, venous congestion, and meningeal haemorrhages due to endothelium damage, as observed in severe feline gurltiosis [10]. Clinical manifestations can include chronic symmetrical or asymmetrical pelvic limb ataxia, ambulatory paraparesis, uni- or bi-lateral hyperactive patellar reflexes, proprioceptive deficit of pelvic limbs, pelvic limb muscular atrophy, diarrhoea, weight loss, coprostasis, urinary and faecal incontinence, and death [1,2,3,4,5,6] (Table 1). Neurological signs are typically associated with neuroanatomical lesions observed in post mortem examinations and histopathological specimens within the spinal cord [1,10,11,12]. Recently, histological and immunohistochemical characterization of vascular alterations in naturally *G. paralysans-*infected domestic cats of Chile unveiled suppurative vasculitis, haemorrhages, vascular congestion, and varicosis of not only spinal cord but also cerebrum-, cerebellum- and brain stem-associated veins [13], thereby supporting endothelium-derived pro-inflammatory innate immune reactions.

## 2. Hypothetical Life Cycle of *Gurltia paralysans* (Angiostrongylidae)

Small felids of the genera *Felis* and *Leopardus* are considered as DH of *G. paralysans*, as sexual replication occurs within these carnivorous mammals. As such, intravascular gravid *G. paralysans* females passing non-embryonated eggs (i.e., those containing 16 blastomeres) within the subarachnoid leptomeningeal veins of spinal cords from domestic cats (*F. catus*) [12], kodkods (*L. guigna*) [1,2,3], margays (*L. wiedii*) and northern tiger cats (*L. tigrinus*) [4] have been reported. Within Chilean territories, other small wild felids of the genus *Leopardus* have been discussed as potential DH [1,2,6], such as the Andean cat (*Leopardus jacobita*) and the Pampas cat (*Leopardus colocolo*). In semiarid, subtropical and tropical regions of South America, the southern tiger cat (*Leopardus guttulus*), Geoffroy’s cat (*Leopardus geoffroyi*) and the ocelot (*Leopardus pardalis*) might also act as DH [1,2,3,4,22]. Moreover, in South, Central and North America, larger wild felids such as the cougar (*P. concolor concolor*), the jaguar (*P. onca*) and the jaguarundi (*H. yagouarundi*) are proposed as DH, but this needs further clarification [1,7]. 

Unfortunately, nothing is known on other aspects of the life cycle, such as the mode of DH infection, exogenous larval development in obligate intermediate hosts (IH), endogenous in vivo migration of infective L3, pre-patency, patency and post-patency [5,6,12]. Neither eggs nor first-stage larvae (L1) have been detected in faeces, blood, bronchial lavage and/or other body fluids of naturally *G. paralysans*-infected domestic cats, northern tiger cats and margays [1,4,22,23]. 

As reported for other nematodes of the family Angiostrongylidae, terrestrial/aquatic gastropods (snails, semi-slugs and slugs) acting as obligate IH as well as paratenic hosts (PH) have been proposed in the biology of *G. paralysans*. Therefore, larval development occurring through moults from first-stage larvae (L1) to second-stage larvae (L2) and to the final infective third-stage larvae (L3) (Figure 2) has recently been proposed [1,5,6,7]. The suspected PH in this life cycle, such as crustaceans, amphibians, reptiles, rodents and birds, might become infected after ingesting L3-carrying gastropod IH, as reported for closely related *Angiostronglylus* species [24,25]. To elucidate the presence of *G. paralysans* larval stages in obligate gastropod IH, a large-scale epidemiological survey was recently conducted in the southern parts of Chile [26]. In this study, 835 terrestrial gastropods from a previously well-recognized endemic focus surrounding the city of Valdivia, Chile [1,26], were collected, demonstrating that neither PCR, enzymatic digestion nor histological examinations revealed the presence of larvae [1,26]. Collected gastropods included slugs of the families Arionidae, Limacidae and Milacidae as well as snails of the family Helicidae. Nonetheless, neither terrestrial semi-slugs (family Helicarionidae) nor aquatic snails were included in this survey, and thus this needs further investigation [1,7,26]. 

Proposed infection routes for felid DH are either after consumption of *G. paralysans* L3-infected gastropod IH or after consumption of L3-infected PH, as initially proposed by Wolffhügel (1934) [3]. As such, in his article, he referred to the colloquial name of feline gurltiosis used by locals, namely “lizard disease”, highlighting the pivotal role of PH in transmission. Alongside lizards, fish, frogs, toads, newts, snakes, turtles, birds, rodents, planarians, crustaceans, insects and myriapods have also been suspected in the life cycle of *G. paralysans* within South America [1,5,21,26]. Likewise, infective *A. cantonensis*-L3 larvae liberated from dead or living gastropods can survive outside IH for a short time, forming an important source of infection. The L3 larvae of *A. cantonensis* can enter new IH through the process known as intermediasis, which might occur in this life cycle as well, thereby extending the survival strategies of *G. paralysans.* These alternative transmission routes can occur with ease in neotropical South American rainforests, which have the highest biodiversity of protist, invertebrate and vertebrate species in the world [7].

Concerning the endogenous migration of *G. paralysans* L3 in felid DH in vivo, nothing is known so far. Hypothetically, L3 migration could be through the small intestinal mucosa in order to reach the mesenteric veins and/or lymphatic vessels of the abdominal viscera and thereafter via connections of either the azygos or the caval venous system (CVS) with the thoracic, lumbar or sacral intervertebral veins until they reach the vertebral venous plexus (VVP, Figure 2) [1]. The VVP is in direct communication with the cranial venous system, and because no valves exist in either of them, blood might flow cranially or caudally, depending on blood pressure [1,27]. *G. paralysans* could take advantage of the absence of valves in the VVP to reach either the spinal subarachnoid space or even the brain [1,12,28]. Similarly, in spinal schistosomiasis in humans, the dissemination of the parasite occurs via the intestinal veins to the VVP [29]. Spinal schistosomiasis usually involves the lower thoracic and lumbosacral spine, probably because the VVP connects the intra-abdominal veins with those of the lower spine [30]. The presence of fertile male and female nematodes, and gravid females passing eggs within the ventral VVP and basivertebral veins located in the vertebral bodies, were isolated during necropsies, confirming the marked angiotropism of *G. paralysans* in DH [1,12]. Moreover, parasitic localization within the VVP’s venous connections may explain the presence of eggs, L1, pre-adults and adults of *G. paralysans* in distant places, such as the cerebrum, cerebellum and anterior chamber of the eye of infected cats [1,20,27,31]. Nonetheless and in contrast to all *Angiostrongylus* species residing within arterial vessels, *G. paralysans* dwells within venous vessels. Thus, the adaptations of *G. paralysans* to the VVP’s venous connections might be associated not only with abiotic factors of the venous microenvironment, such as hypoxia and CO_2_ concentrations, but may also be linked to physical factors (e.g., temperature, blood flow velocity) and even nutrients, among others [1]. It seems indispensable for future investigations on the migratory pathways of *G. paralysans* to include not only vein tropism but also neuroanatomical localization within the subarachnoid VVP in felids [1]. During the patency period, gravid *G. paralysans* will then release un-embryonated eggs into the leptomeningeal vein system. Intravascular eggs will develop into L1, and hatching of the L1 will occur within the VVP, as demonstrated previously [1,12]. The free-released L1 will then breach the alveolar walls in order to access the bronchioles, bronchia and trachea, and are most likely expelled via faeces into the environment. Consistently, the life cycles of other closely related angio-neurotropic metastrongyloid genera of cervids (*Elaphostrongylus* and *Parelaphostrongylus*) might explain the final localizations of *G. paralysans* in the subarachnoid leptomeningeal veins of felid spinal cord. Likewise, adults of *Elaphostrongylus alces* occur in vessels of the epidural space of the vertebral canal and in the skeletal muscles of moose (*Alces alces*) [32,33]. Similar to *G. paralysans*, *E. alces* has obligate gastropod IH, causing neurological disorders in wild moose populations after ingestion of L3-carrying gastropods [32,33]. Earlier researchers suggested that *E. alces* L3 migrate directly from the gut into the epidural space of the caudal vertebral canal, where development to the adult stages takes place. During endogenous development, *E. alces* nematodes produce severe inflammation of the epidural tissue and spinal nerves [32]. In line with this, development of *Elaphostrongylus rangiferi* also takes place only within the brain and spinal cord vessels of reindeer (*Rangifer tarandus*), with subsequent migration of adult nematodes into the skeletal muscle [34]. In the case of *Parelaphostrongylus tenuis*, also known as “meningeal worm” or “brain worm”, which typically occurs in wild cervids (Cervidae), adult nematodes mate in the blood vessels of deer heads, and gravid females start releasing eggs into the circulatory system [35]. In fact, the release of eggs or larvae into the circulatory blood system might be linked with the appearance of *P. tenuis* larvae in the ventral portion of the anterior eye chamber [36], similar to a recent ophthalmic finding in a *G. paralysans-*infected domestic cat of Tenerife Island, Spain [20].

One of the most peculiar features of some *Angiostrongylus* species (Angiostrongylidae) is their strong neurotropism within warm-blooded DH or accidental/aberrant hosts (AH). As such, infectious *A. cantonensis* L3 must migrate through the central nervous system (CNS), where they develop further, reaching the L5 larval stage in the subarachnoidal space within two weeks post infection [37,38]. This part of the life cycle usually does not produce severe signs in DH (e.g., rats); however, infections in AH commonly result in eosinophilic meningitis, with several clinical scenarios [38], as was also the case for *G. paralysans*-infected domestic cats considered by Wolffhügel (1934) as AH. Clinical manifestations are common between *G. paralysans* and *A. cantonensis* due to the signs observed in infected animals resulting in increased intracranial pressure, neural tissue damage, the hosts’ pro-inflammatory response, congestion, thrombosis, thrombophlebitis, varices and thickening of the affected vessels [1,18,37].

## 3. Epizootiology and Environmental Factors Associated with the Presence of Feline Gurltiosis

Environmental factors have been known in the literature to significantly influence the occurrence and epizootiology of metastrongyloid parasites. Abiotic factors such as temperature, humidity and precipitation, as well as biotic factors such as the presence and physiology of DH, obligate IH and PH all play an important part in the continuation of the life cycle, and are involved in parasite mating, egg production, survival of the L1 stages in excreted felid faeces and the persistence of infective *G. paralysans* L3 in PH and/or AH [1,38].

The importance of climate in understanding the rates of transmission of meningeal metastrongyloid nematodes and the risk of infections is often underappreciated. Consequently, it has been reported that, firstly, winter length and severity can be important determinants of the geographic distribution area of wild felids, a situation that has also been observed in wild cervid populations infected with *Elaphostrongylus* and *Parelaphostrongylus* [32,33,34]. Secondly, the climate in summer, especially the amount of precipitation and the length of the summer, determines: (i) the survival of exogenous L1 stages, and (ii) the survival, abundance and mobility of suitable gastropods acting as obligate IH. Thus, climate determines the density of possible DH and gastropod IH, and, in turn, the rates at which each becomes infected, which means that the odds of an encounter between an infected gastropod and a DH depend on the density and degree of the spatial overlaps of both [1,7,38]. In the same way, the majority of terrestrial gastropod populations respond well to wet climatic conditions. In line with this, humidity (precipitation and dew) determines their reproductive success and survival, as well as their mobility in ground litter and on low vegetation, which could explain the epidemiological distribution and endemic foci for feline gurltiosis in the wet southern parts of Chile, not exclusively in rural areas [1,6,10,11,12,26] but also in urban areas, as recently observed [6]. While a lot of data are available on the geographic expansion of *Angiostrongylus* spp. infections in various continents, no data are currently accessible for *G. paralysans* infections within South American territories. The same holds true for the range of possible DH, IH, PH or even AH participating in this neurological parasitosis. Therefore, more epidemiological studies on the occurrence of feline gurltiosis not only in domestic cats, but also in free-ranging wild felids, and gastropod IH, PH and AH, are necessary to better understand the further geographic expansion, re-emergence and/or transmission routes involving other suitable DH, IH and PH in South America as well as in Europe. 

The actual geographical distribution of *G. paralysans* corresponds mainly to South America, specifically to areas of Chile, Argentina, Colombia, Brazil and sporadically in Uruguay [1,2,3,4,5,6,7,10,11,12,14,15,16,17,18,19,20,21]. The presence of this nematode has been reported on the Island of Tenerife, Spain [25]. However, the exact prevalence of *G. paralysans* in these regions is still unknown, despite the fact that infections may be underdiagnosed and the prevalence underestimated. Previous reports have suggested that the focus of infection occurs mainly in rural, peri-urban and urban areas of Southern Chile, which has vast temperate forested areas [1,2,3,5,6,10,11,12,19]. The high endemicity of feline gurltiosis to the temperate southern parts of Chile was recently confirmed in a large-scale epidemiological survey of urban cats (*n* = 171) [6]. As such, Barrios et al. [6] reported the presence of *G. paralysans* in clinically unaffected urban feline populations of three southern Chilean cities, highlighting a considerable percentage of infection (54.4%) in the domestic cats analyzed, indicating, at the same time, that the infection levels seem to vary according to the geographical region [1,2,3,5,6,10,11,12,19]. Nevertheless, no other large-scale epidemiological study has been performed so far to address the occurrence of *G. paralysans* infections in domestic cat populations in any other country.

## 4. Diagnosis

The intra vitam diagnosis of feline gurltiosis is difficult and will depend on the clinical manifestations and diagnosis excluding other myelopathies. Currently, a presumptive diagnosis of *G. paralysans* infection is based on the medical history of affected cats with progressive clinical neurological features originating from potentially endemic areas. A definitive diagnosis can only be performed through a post mortem examination demonstrating and morphologically identifying nematodes in the spinal cord vasculature [1,12,17] (Figure 3). Male specimens of *G. paralysans* have a body length of 12–18 mm and are 0.026–0.032 mm wide in the cephalic region. Females have a body length of 20.5–36.06 mm and a width of 0.082–0.088 mm just anterior to the vulva [1,12,39].

Generally, there are five categories to explore from the appearance of the first clinical manifestations until progressive deterioration of the patients. The first involves physical evaluation of the patient, when it is possible to observe clinical neurological signs related with feline gurltiosis such as chronic symmetrical or asymmetrical pelvic limb ataxia, ambulatory paraparesis, uni- or bilateral hyperactive spinal reflexes, proprioceptive deficit of the pelvic limbs, pelvic limb muscular atrophy, diarrhoea, weight loss, and urinary and faecal incontinence [1,11,12]. Second, laboratory examinations are carried out in order to exclude some levels of anaemia, eosinophilia and thrombocytopenia [17], and even some alterations in the cerebrospinal fluid such as mononuclear pleocytosis [17]. Third, imaging studies are used to explore lesions in the thoracolumbar, lumbar or sacral regions and/or any other typical spinal pathology suggesting diffuse inflammatory spinal cord lesions resulting in severe myelitis or meningomyelitis [1,10,11,12]. Radiography, myelography, computed tomographic myelography (myelo-CT) and magnetic resonance imaging (MRI) are the tools used at present [1,10,17]. As the fourth category, more recently, molecular detection of *G. paralysans* DNA by semi-nested PCR in cerebrospinal fluid, serum and blood samples, using specific oligonucleotides to identify the parasite’s presence [1,39,40]. No other methods are available for a precise ante mortem detection of this nematode. An early diagnosis with conclusive results is required, since laboratory and imaging findings are not sufficient; for that reason, the implementation of molecular tools such as the abovementioned PCR technique may be considered as a complementary and routine diagnostic test for ante mortem detection of feline gurltiosis. Unfortunately, necropsies are conducted exclusively in cases of chronic paraplegia with the prior consent of the owners. The entire spinal cord is then removed from the vertebral canal and incisions are made in the dura mater to expose the affected veins in the subarachnoid space of the cervical, thoracic, lumbar and sacral regions, where the parasites are finally removed and identified (Figure 4A).

Recently, a commercial serological test for canine angiostrongylosis was applied in felines with chronic paraparesis or severe paraplegia [5]. Although the serological test was originally designed for detection of closely related *A. vasorum*, the detection of positive samples (*n* = 7) was achieved; four of the animals were necropsied and subsequently presented findings compatible with feline gurltiosis, both due to the presence of adult nematodes and the histological findings. Collectively, these results suggest cross-reactions between *A. vasorum*-specific antigens and *G. paralysans* when using the Angio Detect TM^®^ Test (IDEXX), suggesting its use as a new diagnostic method for feline gurltiosis in live domestic felines; however, more data on the sensitivity, specificity and characterization of antigens are required for routine intra vitam immunodiagnosis [1,5].

## 5. Clinical Signs

The most frequent clinical manifestation of feline gurltiosis is chronic and progressive ambulatory paraplegia [11,14,15,16,17,18,19,20,21,39,40] (Table 1). The duration of clinical signs ranges from 2 weeks to 48 months [10,14,15,16,17,18,19,20,21,39,40]. Other clinical signs include pelvic limb ataxia, pelvic limb proprioceptive deficit, pelvic limb tremor, pelvic limb muscle atrophy, tail trembling, tail atony, and faecal and urinary incontinence [5,10,11,12,14,15,16,17,18,19,20,21] (Table 1). The neurological signs are associated with the neuroanatomical lesions observed during necropsy and in histopathological specimens [1,18].

The associated haematological abnormalities include non-regenerative anaemia and low mean corpuscular haemoglobin concentrations (hypochromia) [17], indicating chronic inflammatory disease or chronic blood loss. Although eosinophilia has commonly been associated with metazoan parasitism (helminths) in domestic animals and humans, it is not a common finding in cats suffering from feline gurltiosis [17], and has also been reported in dogs with neural angiostrongylosis [41]. Unlike canine angiostrongylosis [41], no signs of coagulopathies with bleedings have been observed in naturally infected cats. However, high levels of urea in the blood have been reported, probably arising from neurogenic urinary dysfunction [17]. A bronchial lavage analysis of five naturally *G. paralysans*-infected cats showed the absence of larval stages and eggs [23]. Ocular lesions (uveitis, chorioretinitis, posterior synechiae, corneal oedema) have recently been reported to be associated with the presence of a motile adult specimen of *G. paralysans* in the anterior chamber of the eye in a domestic cat [20].

## 6. Post Mortem Spinal Cord Examination 

Post mortem macroscopic spinal cord findings in domestic cats with feline gurltiosis are characterized by diffuse submeningeal congestion (Figure 4A) [10,11,12]. The presence of nematode larvae and pre-adult stages can be identified histologically in the subarachnoid veins of the spinal cord, associated with congestion, thrombosis, and thickening of the meningeal vessels (Figure 4B) [10,11,12,17,18]. The presence of suppurative thrombophlebitis and phlebosclerosis with mild smooth-muscle hypertrophy, moderate adventitial fibroplasia and marked subintimal fibrosis of the spinal cord venules have been reported [13,18]. The presence of perivenous fibrosis is considered evidence of chronic inflammatory stages [13]. In some specimens, concentric thickening of the venule wall may produce vascular stenosis [10,18]. Intraluminal papillary projections, with an arboriform appearance, to the interior of dilated venules, have been interpreted as varicose lesions (venular varices) [12,18]. Dilated and tortuous varicose subarachnoid venules may contain thrombi with various levels of organization and *G. paralysans* eggs at different stages of development (Figure 4B,C) [18]. The spinal cord parenchyma may show multiple haemorrhages and extensive foci of malacia, with gitter cells and adjacent gliosis [10,18]. Lymphocytes intermingled with fewer macrophages primarily infiltrate the meninges, forming a perivascular pattern. Mature eosinophils scattered randomly within the leptomeninges have also been observed, which are consistent with extensive spinal leptomeningitis and thrombophlebitis [12,18]. Some animals may also show granulomatous leptomeningitis or suppurative leptomeningitis [18]. White-matter lesions in the spinal cord segments may have variable degrees of Wallerian degeneration, characterized by distension of the myelin sheath diameter, irregular axons, axonal swelling, bulbous axonal fragmentation (caused by the presence of axonal spheroids), microcavitation and focal areas of mineralization [10,12,42]. Varicose venules can also be observed in the white matter of the spinal cord but are associated with recesses in meninges [13,42]. The activation of glial and endothelial cells and immune cell infiltration, visualized with immunohistochemical markers (e.g., GFAP, CNPase, factor VIII, CD3 and CD45R) in affected spinal cord samples indicate gliosis and chronic inflammatory spinal cord lesions subsequent to the ischemia caused by parasitic vascular injury [42]. The predominant cellular infiltrate in the affected spinal cord is of the mononuclear type, indicating the chronic nature of these lesions [42]. A recent study analyzed the presence of histopathological lesions in the cerebrum, cerebellum, and brain stem in 13 feline patients with post mortem spinal lesions due to *G. paralysans* [43]. Congestion and hyperaemia were observed in the peripheral blood vessels of both the dorsal and ventral zones of the cerebrum (*n =* 13). In seven cases, mononuclear cell infiltrates were observed around the choroid plexus, the third and fourth ventricles, and the associated blood vein vessels. Six cats showed thickening of the meninges and two showed perivascular neutrophilic inflammatory infiltrate at the level of the cerebral subarachnoid space [31]. Similar findings of leptomeningeal vascular congestion, varices and perivascular cellular infiltrate were observed in the brains (frontal, temporal and occipital cortices) of 11 *G. paralysans*-infected cats in Brazil [21]. However, no clinical cases of feline gurltiosis have been observed with clinical cerebral, cerebellar, or brain stem syndromes.

Histological samples from 10 *G. paralysans-*infected cats were analyzed at the hepatic level. All samples showed signs of periportal hepatic degeneration and periportal inflammatory infiltration, comprising neutrophilic and mononuclear infiltration, indicating direct injury to the liver [44]. However, the possible mechanisms by which intravascular parasites could cause this type of hepatic injury pattern in the liver and other extraneural tissues remains unclear (Figure 5). *A. vasorum* can induce moderate liver parenchymal parasitic hepatitis and lesions such as interstitial haemorrhages with disseminated inflammatory cells in the portobiliar space or around the centrolobular veins, which was attributed to larval nematode migration [44,45]. In kidneys, feline gurltiosis has been associated with the presence of hyaline protein deposits inside Bowman’s capsule (in 8 out of 10 cases), the thickening of Bowman’s capsule in five cases and the presence of interstitial inflammatory infiltrate, consisting of polymorphonuclear neutrophils (PMN) and eosinophils (in 4 out of 10 cases) [44]. These findings are compatible with glomerulonephritis, which might result from immune-mediated reactions such as the deposition of immune complexes of the host [45].

## 7. Proposed *Gurltia paralysans*-Induced Endothelium and Polymorphonuclear Neutrophil (PMN) Activation, Resulting in Vascular Pathogenesis

Due to its localization in the subarachnoid leptomeningeal veins in vivo, *G. paralysans* is permanently exposed to the feline innate immune system, mainly composed of circulating PMN, other leukocytes (i.e., monocytes, eosinophils and basophils), the vascular endothelium, complement factors, antimicrobial peptides, cytokines and chemokines, among others. Feline PMN are the most abundant leukocyte population in the blood, representing the first line of defense. PMN are recruited immediately after parasite invasion, exhibiting diverse effector mechanisms including production of reactive oxygen species (ROS), degranulation of immunomodulatory molecules, phagocytosis and release of neutrophil extracellular traps (NETs) [46,47,48]. NETs are delicate extracellular fibers mainly composed of decondensed DNA through PAD4-mediated citrullination of global histones (i.e., H1, H2A/H2B, H3 and H4), decorated with antimicrobial/antiparasitic components such as lactoferrin, pentraxin (PTX), calprotein, neutrophil elastase (NE), LL37, myeloperoxidase (MPO), proteinase 3 and cathepsin G [31,41]. Different phenotypes of NETs against protozoan and metazoan parasites have been reported in the literature so far: spread NETs (*spr*NETs), diffuse NETs (*diff*NETs), aggregated NETs (*agg*NETs), and cell-free and anchored NETs [48,49,50,51,52,53,54,55,56,57]. In addition to PMN, endothelial cells are considered highly immunoreactive and, as such, rapidly produce a broad range of molecules, such as adhesion molecules (E-selectin, P-selectin, VCAM-1, ICAM-1, JAM-A), cytokines and chemokines upon activation, thereby inducing pro-inflammatory responses [48]. 

Interestingly, tight interactions between activated endothelium and parasite-induced NETs [54], and also adverse effects of parasite-triggered NETs on endothelial integrity have recently been described [57]. In line with this, within the family Angiostrongylidae, *A. vasorum*-induced activation of primary canine endothelial cells and PMN has recently been published [9]. As such, primary canine aortic endothelial cells (CAEC) as well as PMN responded rapidly to *A. vasorum* as well as soluble antigens (*Av*Ag), showing the pivotal role of these early host innate immune reactions in the pathogenesis of canine angiostrongylosis [9]. Detailed scanning electron microscopy (SEM) analysis unveiled that canine PMN exposed to live *A. vasorum* L3 induced the formation of thick and fine NETs originating from dead PMN, resulting in suicidal NETosis [9]. Concerning *A. vasorum*-induced NET phenotypes, fine *spr*NETs as well as robust *agg*NETs were detected. CAEC-derived pro-inflammatory reactions, shown by the increased expression of various adhesion molecules (E- and P-selectin, VCAM1 and ICAM-1), were also confirmed after stimulation with soluble *Av*Ag but with high individual donor variations [9]. The observed individual variations of endothelium-derived reactions might be linked to different clinical manifestations during canine angiostrongylosis [9]. Metastrongyloid-induced NETosis as well as endothelium reactions are highly conserved within the animal kingdom [40,51], and these immune reactions might be expected in feline gurltiosis as well. Subsequently, chronic vascular inflammation in the course of feline gurltiosis, e.g., thrombophlebitis, haemorrhages and suppurative vasculitis, might be associated with uncontrolled NET formation and endothelium activation in *G. paralysans*-parasitized leptomeningeal vasculature [12]. Likewise, host innate immune-mediated inflammation and vasculature damage have been reported in *A. vasorum*-infected dogs [58,59]. Zoonotically relevant *A. cantonensis* induced an increase in the blood–brain barrier via matrix metalloproteinase 9 (MMP9) upregulation, thereby suggesting parasite-mediated vascular endothelium activation [60]. Activated endothelial cells not only induce leukocyte recruitment and PMN adhesion but also secrete von Willebrand factor (vWF), a large multifunctional glycoprotein with adhesive properties mediating thrombocyte adhesion at the site of vascular damage and inducing thrombosis [59]. Consistently, as indirect evidence of vascular endothelium activation, increased vWF levels have been reported in the circulating blood of *A. vasorum*-infected dogs [61,62]. Interestingly, the *A. vasorum*-derived excretory/secretory and surface proteome revealed various putative modulators of the host’s coagulation system, such as vWF-type D domain protein orthologues, several serine type proteases, protease inhibitors and proteosome subunits that might be involved in coagulopathies [59]. Moreover, several studies on canine angiostrongylosis reported the pathohistological findings associated with long-term alteration of the endothelium physiology and integrity [63,64,65,66]. Consequently, in vivo evidence of endothelium activation demonstrated via immunohistochemical markers (e. g. GFAP, CNPase and Factor VIII) in *G. paralysans*-affected spinal cord vessels has recently been reported [1].

The complex composition of still unknown *G. paralysans*-derived excretory/secretory proteins (ESP), serine type proteases, antigens, inhibitors and/or modulators might lead to endothelium activation as well as intravascular NETosis, thereby causing thrombophlebitis and vasculitis [1,6,12,57]. Thus, it would be of interest to continue investigations into *G. paralysans*-triggered endothelial cell activation or damage of parasitized vessels [12,52] and further to verify the presence of *G. paralysans*-mediated NETosis, as reported for angiotropic *A. vasorum* and *Dirofilaria immitis* [52,54].

## 8. Conclusions

*Gurltia paralysans* is a metastrongyloid nematode that may affect the health status of domestic and wild felids of the genus *Leopardus* and possibly other felids, mainly due to the presence of adult, pre-adult, and larval stages in the subarachnoid space of spinal cord veins, triggering pro-inflammatory reactions resulting in thrombi and venous congestion that eventually become myelopathies with results that practically destroy the quality of life of the affected DH, with sometimes fatal outcomes. A definitive intra vitam diagnosis remains challenging, but the disease can be tentatively identified by clinical signs, blood or cerebrospinal fluid analysis, imaging studies, and molecular tools. Unfortunately, the definitive diagnosis is still post mortem examination based on the presence of *G. paralysan*s within the spinal cord veins. As feline gurltiosis can also affect wild felids, biologists and ecologists involved in conservation programs of endangered species in South, Central, and North America should be aware of this neglected neuro-angiotropic nematode and avoid possible spillovers resulting from the presence of feral cats in national parks or protected biomes. Finally, veterinarians and cat owners should be warned and aware of the environmental control of obligate IH and/or PH as an efficient way to reduce the infection ratio and, therefore, the magnitude of the damage. 

## Figures and Tables

**Figure 1 pathogens-11-00792-f001:**
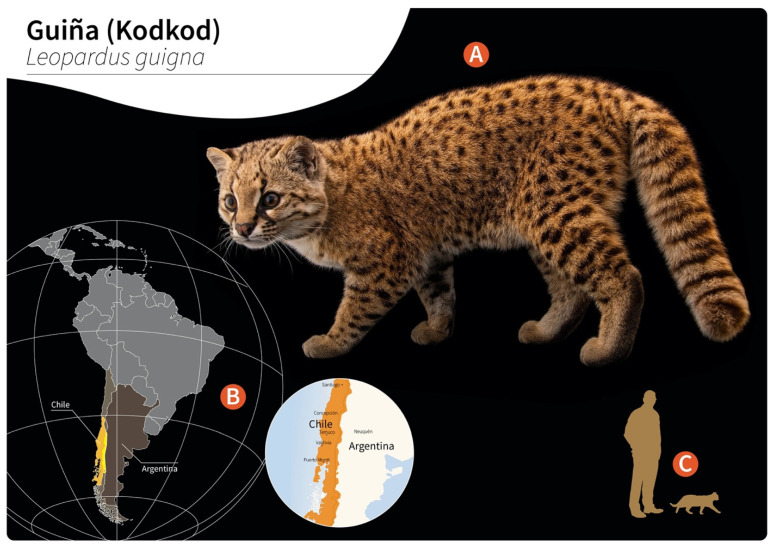
Distribution of wild guiñas (syn. huiñas, kodkods, spotted tiger cat) in South America. (**A**) Adult specimen of a guiña (*Leopardus guigna*) (image reprinted with permission from © Joel Sartore/Photo Ark, 2022). (**B**) Geographic distribution of guiña in Chile (orange) and Argentina (yellow). (**C**) Scale representation of an adult guiña.

**Figure 2 pathogens-11-00792-f002:**
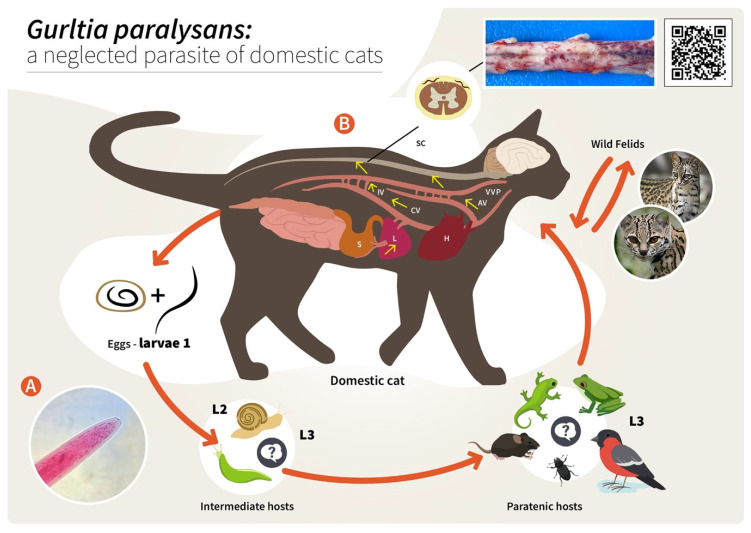
Proposed life cycle and migration pathways of *Gurltia paralysans*. (**A**) Cranial end of an adult specimen of *G. paralysans*. (**B**) Domestic cats (*Felis catus*) or wild felids (*Leopardus* spp.) acquire the L3 larvae by ingesting an infected obligate intermediate host (gastropods) or paratenic hosts (lizards, rodents, amphibians, birds or insects). Infective larvae penetrate the stomach and enter the hepatic portal system, and then the caudal vena cava and/or the azygous venous system. From these vein systems, the larvae migrate to the spinal cord via the intervertebral veins and the vertebral venous plexus. The larvae invade the veins of the subarachnoid space of the spinal cord, where they mature and lay eggs. It is still unknown on how domestic cats eliminate the eggs or the first-stage larvae (L1) into the environment, their further development into the L2 and L3 larval stages, or how the obligate intermediate hosts become infected with L1. AV: azygos vein; CV: caudal vena cava; IV: intervertebral veins; H: heart; L: liver; S: stomach; SC: spinal cord; VVP: vertebral venous plexus; L1: first-stage larvae; L2: second-stage larvae; L3: third-stage larvae. The inserted QR code shows a video of a *G. paralysans*-infected cat with clinical signs of paraparesis.

**Figure 3 pathogens-11-00792-f003:**
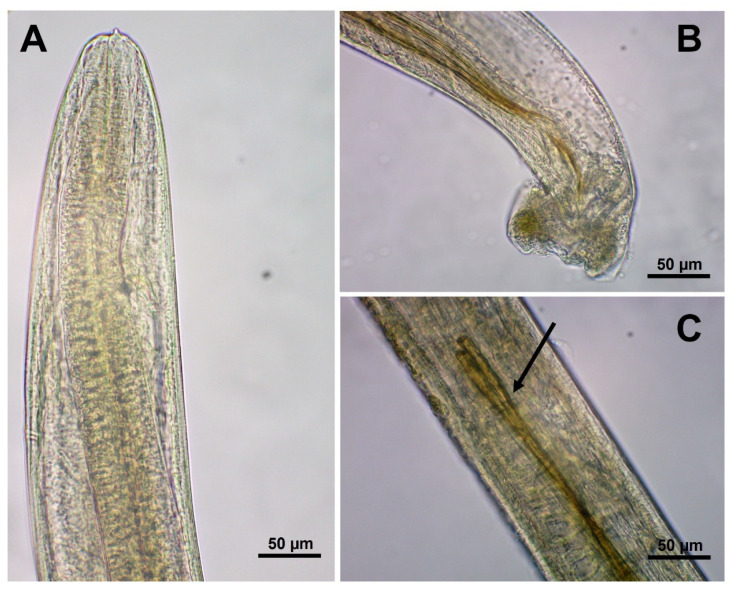
Microscopic view of *Gurltia paralysans*. (**A**) Cephalic end of the specimen showing a tooth at the anterior margin (scale bar: 50 µm). (**B**) Caudal end of a male, showing the small copulatory bursa (scale bar: 50 µm). (**C**) Higher magnification of a male caudal extremity, showing the spicules (arrow) (scale bar: 50 µm).

**Figure 4 pathogens-11-00792-f004:**
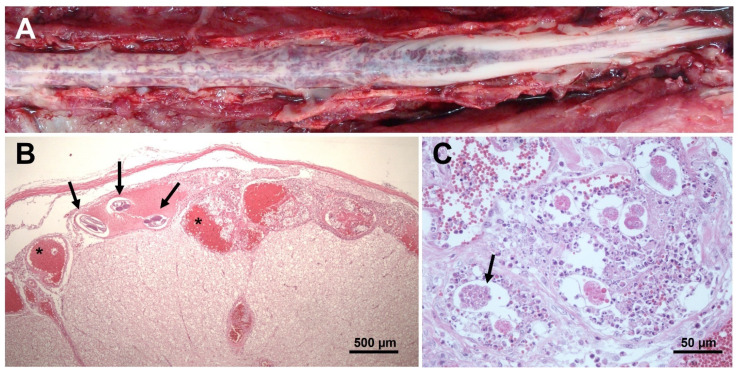
Macroscopic and microscopic lesions of the spinal cord in a *Gurltia paralysans*-infected cat. (**A**) Lumbar, sacral and caudal segments of the spinal cord showing severe and diffuse submeningeal vascular congestion. (**B**) Histopathological view of transverse sections of an adult of *G. paralysans* (arrows) inside a subarachnoid vein and vascular congestion (asterisk) in the spinal subarachnoid space; HE, 4 × (scale bar: 500 µm). (**C**) Histopathological section of the spinal cord parenchyma showing developing eggs of *G. paralysans* (arrow); HE, 40 × (scale bar: 50 µm).

**Figure 5 pathogens-11-00792-f005:**
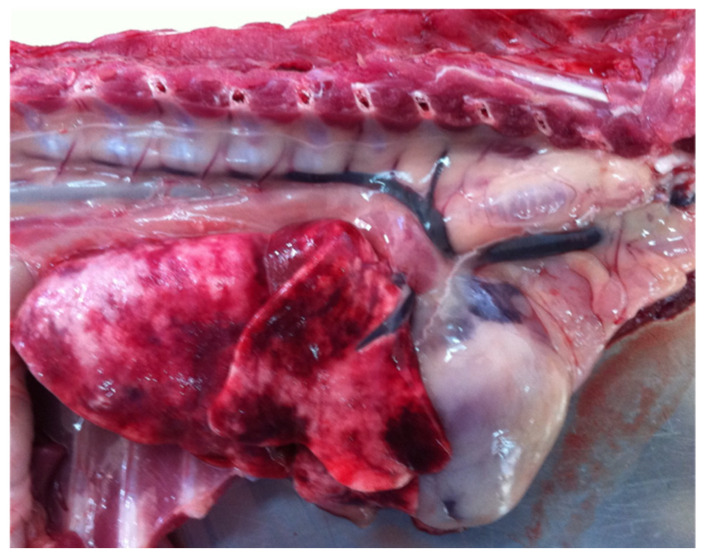
Macroscopic view of the right lung with multiple haemorrhagic foci in a cat with feline gurltiosis.

**Table 1 pathogens-11-00792-t001:** Reported cases of feline gurltiosis including age, geographic location, clinical presentation and diagnosis.

Age	Location	Number of Cases	Clinical Presentation	Diagnosis	Reference
1–3 y	Chile (Los Ríos/Los Lagos regions)	3	Paraparesis (ambulatory) PL ataxia PL muscle atrophy Anal/urinary incontinence	Post mortem (histopathology)	[12]
6–8 m	Colombia (Antioquia municipality)	6	Paraparesis (ambulatory) PL ataxia Spinal pain PL muscle atrophy Anal/urinary incontinence Decrease superficial/deep pain in PL	Post mortem (histopathology, Myelo)	[14]
2 y	Argentina (Buenos Aires province)	1	Paraparesis (non-ambulatory) PL muscle atrophy Increase spinal reflexes in PL Decrease superficial/deep pain in PL	Post mortem (histopathology)	[15]
NA	Uruguay (Fray Bentos)	2	Paraparesis (ambulatory) Paraplegia PL ataxia	Post mortem (histopathology)	[16]
1–3 y	Chile	3	Paraparesis (ambulatory) PL ataxia PL muscle atrophy Anal/urinary incontinence Spinal hyperaesthesia PL trembling Increase spinal reflexes in PL Paraplegia	Post mortem (histopathology, specimens extracted from SSE)	[11]
8 m–10 y	Chile	9	Paraparesis (ambulatory) Paraparesis (non-ambulatory) Paraplegia PL ataxia Increase spinal reflexes in PL Spinal hyperaesthesia Anal/urinary incontinence	Post mortem (histopathology, specimens extracted from SSE, Myelo, CT, MRI	[17]
NA	Brazil (Río Grande do Sul)	4	Paraparesis (ambulatory) PL muscle atrophy Vesical atony Tail atony	Post mortem (histopathology)	[18]
NA	Argentina (Santa Fé)	3	Paraparesis (ambulatory) Paraplegia Decrease spinal reflexes in PL Decrease superficial pain in PL Skin lesions in the metatarsal region	Post mortem (histopathology)	[19]
8 m	Chile (Ancud, Los Lagos regions)	1	Paraparesis (ambulatory) Anal/urinary incontinence Tail atony	Myelo-CT, CSF (mononuclear pleocytosis), post mortem (histopathology)	[10]
2 y	Spain (Tenerife)	1	Uveitis in left eye	Specimen extracted from anterior chamber of the eye, PCR	[20]
NA	Brazil (Pernambuco)	11	Paraparesis (ambulatory) PL ataxia, PL muscle atrophy Skin lesions in metatarsal and phalangeal regions	Post mortem (histopathology)	[21]
36 m	Chile	10	Paraparesis (ambulatory) Paraplegia PL ataxia Anal/urinary incontinence	Post mortem (histopathology), IDEXX (Angio Detect), specimens extracted from SSE	[5]

NA: not available; SSE: spinal subarachnoid space; PL: pelvic limbs; Myelo: myelography, My—lo-CT: computed tomography myelography; MRI: magnetic resonance imaging; CSF: cerebrospinal fluid; y: years; m: months.

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
