# Peer review of "Gurltia paralysans: A Neglected Angio-Neurotropic Parasite of Domestic Cats (Felis catus) and Free-Ranging Wild Felids (Leopardus spp.) in South America"

_pathogens, 2022, doi:10.3390/pathogens11070792_

Round 1
Reviewer 1 Report
A very interesting review on Gurltia paralysans in cats.
Just few misspelling:
Line 184: Elaphostrongylus instead of Elaphostromgylus
Line 197: post infection instead of post infectionem
As a metastrongylid, larvae L1 would most likely be shed in faeces (Line 170-174). Why coprology is not suggested as a diagnostic tool?.
Author Response
REVIEWER 1 RESPONSE
Dear Reviewer
Thanks for your comments.
Comments and Suggestions for Authors
A very interesting review on Gurltia paralysans in cats.
Just few misspelling:
Line 184: Elaphostrongylus instead of Elaphostromgylus
Done, it was changed
Line 197: post infection instead of post infectionem
Done, it was changed
As a metastrongylid, larvae L1 would most likely be shed in faeces (Line 170-174). Why coprology is not suggested as a diagnostic tool?.
It was added “ Neither eggs nor first-stage larvae (L1) have been detected in faeces, blood, bronchial lavage and/or other body fluids of naturally G. paralysans-infected domestic cats, northern tiger cats and margays [1,4,14,15].” (Page 3, Line 222-224)
Reviewer 2 Report
The manuscript by Rojas-Barón et al. deals with a review about Gurltia paralysans, a parasite causing severe neurological pathology in domestic and wild felids in South America. The paper is interesting, as it reviews the few information available on this parasite.
However, several major changes are required before deserving publication.
First, there is no information on the size of adult parasites and this is important for pathologists, who must confirm or rule out the clinical suspicion during post mortem examination. Is it easy to visualize them during post mortem examination?Second, the text should be carefully checked for several errors and imprecisions throughout the whole manuscript; also English language should be revised.
Some requested changes are listed below
Line 20: change “Leopardus guinia” to “Leopardus guigna”
Lines 31-33: the sentence should be rephrased as “post mortem necropsies” is opposite to intra vitam diagnosis. Moreover it is better to use “post mortem examinations” instead of “post mortem necropsies”
Line 93: change to “An adult of Guigna”
Line 123: add “the” between “elucidate” and “presence”
Line 128: add “the” between “revealed” and “presence”
Line 197: change “post infectionem” to “post infection”
Line 207: change “Felis domesticus” to “Felis catus”
Lines 273 e 489: change “post-mortem” to “post mortem”
Line 275: What is the size of adult parasites? Is it easy to visualize them during the post mortem examination?Line 376: it is better to use “brains” instead of “encephalons”
Line 400: change “congestive vasculature” to “vascular congestion”
Line 401: delete “specimen” and change “congestive vessels” to “vascular congestion”
Line 406: change “haemorrhagies foci” to “haemorrhagic foci”
Line 420: add “the” between “representing” and “first”
Line 482: the part of sentence “domestic and wild felids of the genus Leopardus” should be rewritten, as it seems to indicate there are domestic species of the genus Leopardus.
Author Response
REVIEWER 2 RESPONSE
Dear Reviewer
Thanks for your comments.
Comments and Suggestions for Authors
The manuscript by Rojas-Barón et al. deals with a review about Gurltia paralysans, a parasite causing severe neurological pathology in domestic and wild felids in South America. The paper is interesting, as it reviews the few information available on this parasite.
However, several major changes are required before deserving publication.
First, there is no information on the size of adult parasites and this is important for pathologists, who must confirm or rule out the clinical suspicion during post mortem examination. Is it easy to visualize them during post mortem examination?
Done, it was added “Male specimens of G. paralysans have a body length of 12–18 mm and are 0.026–0.032 mm wide in the cephalic region. Females have a body length of 20.5–36.06 mm and a width of 0.082–0.088 mm just anterior to the vulva [1]. (Page 7, lines 293-295)
Second, the text should be carefully checked for several errors and imprecisions throughout the whole manuscript; also English language should be revised.
Done, the manuscript undergone English language editing by MDPI
Some requested changes are listed below
Line 20: change “Leopardus guinia” to “Leopardus guigna”
Done, it was changed
Lines 31-33: the sentence should be rephrased as “post mortem necropsies” is opposite to intra vitam diagnosis. Moreover it is better to use “post mortem examinations” instead of “post mortem necropsies”
Done, it was changed “Diagnosis is based on clinical neurological signs, imaging findings through computed tomography (CT), myelography, magnetic resonance imaging (MRI) and post mortem examination.”
Line 93: change to “An adult of Guigna”
Done, it was changed
Line 123: add “the” between “elucidate” and “presence”
Done, it was changed
Line 128: add “the” between “revealed” and “presence”
Done, it was changed
Line 197: change “post infectionem” to “post infection”
Done, it was changed
Line 207: change “Felis domesticus” to “Felis catus”
Done, it was changed
Lines 273 e 489: change “post-mortem” to “post mortem”
Done, it was changed
Line 275: What is the size of adult parasites? Is it easy to visualize them during the post mortem examination?
Done, it was added “Male specimens of G. paralysans have a body length of 12–18 mm and are 0.026–0.032 mm wide in the cephalic region. Females have a body length of 20.5–36.06 mm and a width of 0.082–0.088 mm just anterior to the vulva [1]. (Page 7, lines 293-297).
Line 376: it is better to use “brains” instead of “encephalons”
Done, it was changed
Line 400: change “congestive vasculature” to “vascular congestion”
Done, it was changed
Line 401: delete “specimen” and change “congestive vessels” to “vascular congestion”
Done, it was changed
Line 406: change “haemorrhagies foci” to “haemorrhagic foci”
Done, it was changed
Line 420: add “the” between “representing” and “first”
Done, it was changed
Line 482: the part of sentence “domestic and wild felids of the genus Leopardus” should be rewritten, as it seems to indicate there are domestic species of the genus Leopardus.
Done, it was added “…Gurltia paralysans is a metastrongyloid nematode that may affect the health status of domestic and wild felids of the genus Leopardus and possibly other felids, mainly due to the presence of adult…”
Reviewer 3 Report
The review Gurltia paralysans: a neglected angio-neurotropic parasite of domestic cats (Felis catus) and free-ranging wild felids (Leopardus spp.) in South America is well documented, written and structured. I read the manuscript with great interest.
The manuscript gathers together very useful information regarding the epidemiology of the parasite, the diagnosis, the clinical and morphopatologicalc lessions, immune response. Also, the authors highlight very clearly what are the current limitations regarding the biological cycle of the parasite and intra-vitam diagnosis.
I consider that the manuscript could be accepted for publication.
Author Response
Dear Reviewer
Thanks for your comments.
Regards